# $\zeta$-*mixup*: Richer, More Realistic Mixing of Multiple Images

**Kumar Abhishek**[1]                                         KABHISHE@SFU.CA
**Colin J. Brown**[2]                          COLIN.BROWN@HINGEHEALTH.COM
**Ghassan Hamarneh**[1]                                   HAMARNEH@SFU.CA
[1] *School of Computing Science, Simon Fraser University, Canada*
[2] *Hinge Health, Canada*

**Editors:** Accepted for publication at MIDL 2023

## Abstract

Data augmentation (DA), an effective regularization technique, generates training samples to enhance the diversity of data and the richness of label information for training modern deep learning models. *mixup*, a popular recent DA method, augments training datasets with convex combinations of original samples pairs, but can generate undesirable samples, with data being sampled off the manifold and with incorrect labels. In this work, we propose $\zeta$-*mixup*, a generalization of *mixup* with provably and demonstrably desirable properties that allows for convex combinations of $N \geq 2$ samples, thus leading to more realistic and diverse outputs that incorporate information from $N$ original samples using a $p$-series interpolant. We show that, compared to *mixup*, $\zeta$-*mixup* better preserves the intrinsic dimensionality of the original datasets, a desirable property for training generalizable models, and is at least as fast as *mixup*. Evaluation on several natural and medical image datasets shows that $\zeta$-*mixup* outperforms *mixup*, CutMix, and traditional DA methods.

**Keywords:** data augmentation, mixup, intrinsic dimensionality, data manifold

## 1. Introduction

Given the large parameter space of deep learning models, training on small datasets tends to cause the models to overfit to the training samples, which is especially a problem when training with data from high dimensional input spaces such as images, and consequently, benefits from data augmentation (DA) techniques for improved generalization performance. *mixup* (Zhang et al., 2018), a popular DA method, generates convex combinations of pairs of original training samples and linear interpolations of corresponding labels with a hyper-parameter $\lambda \sim [0, 1]$. The primary hypothesis of *mixup* and many derivatives is that a model should behave linearly between any two training samples, even if the distance between samples is large. This implies that we may train the model with synthetic samples that have very low confidence of realism; in effect, over-regularizing. We instead argue that we should only synthesize examples with high confidence of realism, and that a model should only behave linearly nearby training samples, supported by research in cognitive sciences showing that human perception between object category boundaries is warped and not as linear as *mixup* seems to suggest (Beale and Keil, 1995; Newell and Bülthoff, 2002).

Consider the $\mathcal{K}$-class classification task, where we are provided with a dataset of $m$ points $\{x_i\}_{i=1}^m$ in a $\mathcal{D}$-dimensional ambient space $\mathbb{R}^{\mathcal{D}}$ with the corresponding labels $\{y_i\}_{i=1}^m$ in a label space $\mathcal{L} = \{l_1, \cdots, l_{\mathcal{K}}\} \in \mathbb{R}^{\mathcal{K}}$. Keeping in line with the manifold hypothesis (Cayton, 2005; Fefferman et al., 2016), which states that complex data manifolds in high dimensional

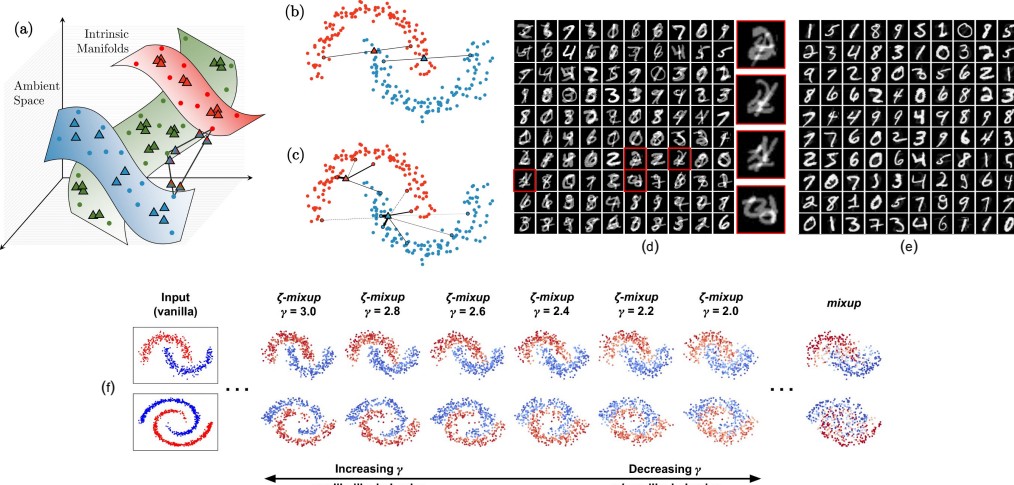

Figure 1: (a) An overview of $\zeta$-*mixup* with original ($\circ$) and synthetic ($\triangle$) samples. Note how *mixup* ((b), (d)) does not respect individual class boundaries and can generate incorrect samples, that lie off the data manifold, with incorrect labels. $\zeta$-*mixup* ((a), (c), (e)) can mix any number of samples (e.g., 3 in (a), 4 or 8 in (c), and 25 in (e)) and the generated samples remain close to the original distribution while incorporating rich information from several samples. (f) The hyperparameter $\gamma$ in $\zeta$-*mixup* formulation can control the diversity of the synthetic samples.

ambient spaces are actually made up of samples from manifolds with low intrinsic dimensionalities ($\mathcal{D}_{\text{int}}$), we assume that the $m$ points are samples from $\mathcal{K}$ manifolds $\{\mathcal{M}_i\}_{i=1}^{\mathcal{K}}$ with $\mathcal{D}_{\text{int}}$ as $\{d_i\}_{i=1}^{\mathcal{K}}$, where $d_i << \mathcal{D} \ \forall i \in [1, \mathcal{K}]$ (Fig. 1 (a)). We seek an augmentation method that facilitates a denser sampling of each intrinsic manifold $\mathcal{M}_i$, thus generating more real and more diverse samples with richer labels. Following Wood et al. (2021); Wood (2021), we consider three criteria for evaluating the quality of synthetic data: **(i) realism:** allowing the generation of correctly labeled synthetic samples close to the original samples, ensuring the realism of the synthetic samples, **(ii) diversity:** facilitating the synthesis of more diverse samples by allowing exploration of the input space, and **(iii) label richness** when generating synthetic samples while still staying on the manifold of realistic samples. Additionally, we aim for: **(iv) valid probabilistic labels** along with **(v) computationally efficient** augmentation of training batches (e.g., avoiding inter-sample distance calculations).

To this end, we propose to synthesize a new sample $(\hat{x}_k, \hat{y}_k)$ as $\hat{x}_k = \sum_{i=1}^{N} w_i x_i; \quad \hat{y}_k = \sum_{i=1}^{N} w_i y_i$, where $w_i$s are the weights assigned to the $N$ samples being mixed. One such suitable weighting scheme is to sample weights from the terms of a $p$-series, i.e., $w_i = i^{-p}$, which is a convergent series for $p \geq 1$. Extending the idea of local synthetic instances for connectome augmentation (Brown et al., 2015), we adopt the following formulation: given $N$ samples (where $2 \leq N \leq m$ and thus, theoretically, the entire dataset), a $N \times N$ random permutation matrix $\pi$, and the resulting randomized ordering of samples $s = \pi[1, 2, \ldots, N]^T$, the weights are defined as $w_i = \frac{s_i^{-\gamma}}{C}, \quad i \in [1, N]$, where the hyperparameter $\gamma$ allows us to control how far the synthetic samples can stray away from the original samples. $C$ is the normalization constant to ensure that $w_i$ must satisfy $w_i \geq 0 \ \forall i$ and $\sum_{i=1}^{N} w_i = 1$, such that $\hat{y}_k$ is a valid probabilistic label, where $C = \sum_{j=1}^{N} j^{-\gamma}$ is the $N$-truncated Riemann

Table 1: Classification error on CIFAR datasets averaged over 3 runs ($\gamma \in \mathrm{U}[\gamma_{\min}, 4.0]$).

| Method | CIFAR-10 ResNet-18 | CIFAR-100 ResNet-18 | Method | CIFAR-10 | | CIFAR-100 | |
|---|---|---|---|---|---|---|---|
| | | | | ResNet-18 | ResNet-50 | ResNet-18 | ResNet-50 |
| ERM | 5.48 | 23.33 | CutMix | 4.13 | 4.08 | 19.97 | 18.99 |
| *mixup* | 4.68 | 21.85 | + *ζ-mixup* | **3.84** | **3.61** | **19.54** | **18.86** |
| *ζ-mixup* | **4.42** | **21.35** | | | | | |

Table 2: Micro-averaged F1 score on skin lesion image datasets ($\gamma = 2.8$).

| Method | ISIC 2016 | | ISIC 2017 | | ISIC 2018 | | DermoFit | |
|---|---|---|---|---|---|---|---|---|
| | ResNet-18 | ResNet-50 | ResNet-18 | ResNet-50 | ResNet-18 | ResNet-50 | ResNet-18 | ResNet-50 |
| ERM | 0.7836 | 0.8127 | 0.7383 | 0.6867 | **0.8756** | 0.8653 | 0.8269 | 0.8500 |
| *mixup* | 0.7968 | 0.8179 | 0.7333 | 0.7433 | 0.8394 | 0.8601 | 0.8577 | 0.8500 |
| *ζ-mixup* | **0.8654** | **0.8602** | **0.7633** | **0.7733** | **0.8756** | **0.9016** | **0.8731** | **0.8962** |

zeta function (Riemann, 1859) $\zeta(z)$ evaluated at $z = \gamma$, and thus we call our method $\zeta$-*mixup*. Since there exist $N!$ possible $N \times N$ random permutation matrices, given $N$ original samples, $\zeta$-*mixup* can synthesize $N!$ new samples for a single $\gamma$, unlike *mixup* which can only synthesize 1 new sample per sample pair for a single $\lambda$. Moreover, as a result of its formulation, $\zeta$-*mixup* presents two desirable properties: **(1)** for all values of $\gamma \geq \gamma_{\min} = 1.72865$, the weight assigned to one sample is greater than the sum of weights assigned to all other samples, implicitly introducing the desired notion of linearity in only the locality of original samples; and **(2)** for $N = 2$ and $\gamma = \log_2\left(\frac{\lambda}{1-\lambda}\right)$, $\zeta$-*mixup* simplifies to *mixup*.

## 2. Results and Discussion

Using a PCA-based local $\mathcal{D}_{\mathrm{int}}$ estimator calculated using a $k$-nearest neighborhood around each sample, with $k = 128$ (Fukunaga and Olsen, 1971), we find that $\mathcal{D}_{\mathrm{int}}$ for CIFAR-10 and CIFAR-100 using $\zeta$-*mixup* are lower than using *mixup*: $26.83 \pm 6.53$ (versus $35.43 \pm 9.47$) and $24.76 \pm 6.22$ (versus $32.41 \pm 8.65$), respectively, thus showing that $\zeta$-*mixup* indeed preserves the low $\mathcal{D}_{\mathrm{int}}$ that natural image datasets lie in (Ruderman, 1994; Pope et al., 2021), while *mixup*'s off-manifold sampling leads to an inflated estimate of local $\mathcal{D}_{\mathrm{int}}$. Tables 1 and 2 show the classification performance using traditional DA techniques, e.g., rotation, flipping, and cropping ("ERM"), against those trained with *mixup* and $\zeta$-*mixup* outputs as well as compare the benefit of applying $\zeta$-*mixup* to an orthogonal DA method, CutMix (Yun et al., 2019), as evaluated on natural: CIFAR-10 and CIFAR-100 and medical (skin lesion): ISIC 2016 (Gutman et al., 2016), 2017 (Codella et al., 2018), and 2018 (Codella et al., 2019), and DermoFit (Ballerini et al., 2013) image datasets. We report the error rate and the micro-averaged F1-score for natural and medical image datasets, respectively, since the latter are class-imbalanced. We observe that $\zeta$-*mixup* improves performance across the board. Our optimized $\zeta$-*mixup* implementation is $2.1\times$ faster than the original *mixup* implementation, while similar training time is recorded for both of them for CIFAR-10/100 ($\sim$ 1h 20m).

**Conclusion:** We proposed $\zeta$-*mixup*, a parameter-free multi-sample generalization of the popular *mixup* technique for data augmentation that combines $N \geq 2$ samples without significant computational overhead. The $\zeta$-*mixup* formulation allows for the weight assigned to one sample to dominate all the others, thus ensuring the synthesized samples are on or close to the original data manifold. This leads to generating samples that are more realistic and, along with allowing $N > 2$, generates more diverse samples with richer labels compared to *mixup*. Future work will include exploring $\zeta$-*mixup* in the learned feature space.

## Acknowledgments

The authors are grateful to StackOverflow user `obchardon` and Ashish Sinha for code optimization suggestions and to Saeid Asgari Taghanaki for initial discussions. The authors are also grateful for the computational resources provided by NVIDIA Corporation and Digital Research Alliance of Canada (formerly Compute Canada). Partial funding for this project was provided by the Natural Sciences and Engineering Research Council of Canada (NSERC).

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
