# OpenReview forum: "$\zeta$-mixup: Richer, More Realistic Mixing of Multiple Images"
_MIDL.io/2023/Short_Paper_Track — MIDL 2023 Short paper track Poster_

### Official Review · Reviewer_1bTm · 2023-04-20
**Interesting improvement over mixup**

**Rating:** 8
**Confidence:** 4

**Review:**

Summary: The paper introduces an improvement over a popular data augmentation method called mixup. The proposed method termed ζ-mixup improves over mixup in two ways 1) the method is able to mix more than two images/samples, and 2) the generated sample is close to one of the mixed samples. The second part ensures that the generated samples are more realistic and a model should behave linearly for closer samples only.

Strength:

The method is well motivated and the idea of using more than two samples for mix-up is interesting. The idea of assigning more weight to one sample is also well-motivated from cognitive sciences.

The method is tested on natural and medical images to demonstrate its applicability.

The paper overall is well-written and easy to understand.

Weakness:

The empirical section could use some experiments. Especially, it would be interesting to see results with different hyperparameter values e.g., number of samples mixed together and different weighting scheme.

It is not clear right now what is helping the proposed method more, the weighting scheme or mixing more than two samples.

---

### Official Review · Reviewer_VHyC · 2023-04-26
**Review of $\zeta$-mixup: Richer, More Realistic Mixing of Multiple Images**

**Rating:** 10
**Confidence:** 4

**Review:**

This paper presents a method for data augmentation, which extends the popular mixup method. mixup works by augmenting with convex combinations of pairs of input data.$\zeta$-mixup extends this by considering convex combinations of $N \geq 2$ training samples. The weights are computed as a $p$-series, $w_i = i^-p$ (normalized to sum to one). The results show improved performance and also that the augmented samples have lower intrinsic dimensionality than those from mixup.

This is an excellent idea, especially the focus on generating more realistic augmentation samples and keeping the intrinsic dimensionality lower.

The results are quite convincing. Although, I wonder if there are more recent improvements of mixup that would make for better state-of-the-art comparisons (I'm not familiar with such follow-up work, so I am not sure).

The paper is very clearly written (especially for a short paper where it is hard to get ideas across concisely). The only point that was not completely clear to me was the motivation for using the $p$-series for weights. Was this solely to make sure that there are varying weights that will stay well-behaved for larger $N$? Or, is there some other property of this series that is desirable?